# Analysis of Well-Being and Anxiety among University Students

**DOI:** 10.3390/ijerph17113874

**Published:** 2020-05-30

**Authors:** Luis Felipe Dias Lopes, Bianca Michels Chaves, Adriane Fabrício, Adriana Porto, Damiana Machado de Almeida, Sandra Leonara Obregon, Mauren Pimentel Lima, Wesley Vieira da Silva, Maria Emilia Camargo, Claudimar Pereira da Veiga, Gilnei Luiz de Moura, Luciana Santos Costa Vieira da Silva, Vânia Medianeira Flores Costa

**Affiliations:** 1Graduate Program in Business Administration—PPGA, Federal University of Santa Maria—UFSM, Av. Roraima n° 1000, Cidade Universitária, Camobi, Santa Maria, Rio Grande do Sul estado 97105-900, Brazil; lflopes67@gmail.com (L.F.D.L.); biancamichelsc@gmail.com (B.M.C.); adrianefabricio@yahoo.com.br (A.F.); adm.damiana@gmail.com (D.M.d.A.); sandraobregon12@gmail.com (S.L.O.); maurenlima@hotmail.com (M.P.L.); mr.gmoura.ufsm@gmail.com (G.L.d.M.); vania.costa@ufsm.br (V.M.F.C.); 2Department of Business Administration, University Luterana of Braszil—ULBRA, Cachoeira do Sul—Rua Martinho Lutero 301, Bairro Universitário, Cachoeira do Sul, Rio Grande do Sul 96.501-595, Brazil; adrianaportoadm@gmail.com; 3Department of Management, Federal Rural University of Semi-Arid—UFERSA, Rua Francisco Mota, 572, Presidente Costa e Silva, Mossoró, RN 59625-900, Brazil; wesvsilva@gmail.com; 4Graduate Program in Industrial Engineering, University of Caxias do Sul, Rua Francisco Getúlio Vargas 1130, Caxias do Sul 95070-560, Brazil; mariaemiliappga@gmail.com; 5School of Management—PPGOLD, Federal University of Parana—UFPR, 632 Prefeito Lothário Meissner Ave, Jardim Botânico, Curitiba, Paraná 80210-170, Brazil; 6Department of Management, Municipal Center of São José, Rua Jaír Viêira, 2–68, Kobrasol, São José, Santa Catarina 88102-180, Brazil; luvcosta10@gmail.com

**Keywords:** public health, well-being, anxiety, behavioral disease, university student

## Abstract

This article aims to interrelate dimensions of the well-being validation instruments proposed by Watson, Clark and Tellegen (PANAS) with generalized anxiety dimensions proposed by Spitzer et al. (GAD-7) and state-trait anxiety inventories proposed by Biaggio and Natalício (IDATE), using partial least squares structural equation modeling (PLS-SEM), in the case of individual university students in southern Brazil and the city of Buenos Aires, Argentina. We conducted a behavioral study, characterized as exploratory-descriptive, by applying a questionnaire survey to collect data though face-to face interviews to a group of 460 university students from June to August 2019. A non-probabilistic sampling method for convenience was used, justified by the heterogeneous incidence of the participants. Our results support most of the proposed hypotheses. Only one hypothesis was rejected, i.e., that the Positive Affection Scale (WBS) is not related to the State Anxiety Inventory (IAE)—when a person is feeling in full activity, this situation does not affect the momentary state, characterized by tension, apprehension and by increased activity in the autonomic nervous system. In terms of the subjective well-being of students, 14.13% were found to have a low rating. 86.74% were found to have generalized anxiety; 75% had trait anxiety, and 80.22% had state anxiety. Our results indicate the need for preventive measures to minimize anxiety and help maintain necessary levels of well-being during this phase of academic development and when forging a professional career. It is expected that new studies will contribute to the advancement of such themes, particularly with university students.

## 1. Introduction

The number of cases of mental illness among university students is considered a public health problem, but these young people often do not seek appropriate treatment. Entry into higher education is also part of the transition to adulthood and can result in an overload of anxiety, fear and challenges, which can cause anxiety disorders (one of the most common disorders). Disorders recorded during this period can lead to use of cigarettes and illicit drugs, insomnia and, consequently, low academic performance. Research shows that self-medication with drugs often minimizes the symptoms of generalized anxiety disorder [1].

Denomination of happiness (on which subjective well-being (SWB) is based) follows from what is known as the hedonic tradition [2]. Sociologists and quality-of-life researchers have had a major influence on this topic by attempting to determine the demographic factors that influence subjective well-being [3,4,5].

In the well-being literature, two main approaches have emerged which have a significant conceptual distinction: one addresses happiness (hedonic well-being), and one addresses human potential (eudaimonic well-being) [6,7]. However, Paschoal and Tamayo [2] affirm that happiness and well-being are terms that end up being treated in a mixed way in the scientific literature and, in general, are used as synonyms.

For Diener et al. [8], several different lines of research have come together in the history of the SWB field, involving scientific analysis of how people live their lives, both in the present moment and over longer periods. These measurements include individuals’ emotional reactions to events, their mood and the judgments they make about realizing and fulfilling life in key areas such as marriage and work. In this way, SWB is concerned with the study of what lay people would call happiness or satisfaction.

Instruments used in SWB analysis include those of Watson et al. [9], which emphasizes the relationship between positive and negative well-being. SWB is one of the three main ways of measuring quality of life in societies, together with economic and social indicators [8,10]. How individuals think and feel about their own life is a vital factor in understanding well-being in any context and takes into account not only the opinion of experts but people in society in general. A high level of SWB is a necessary feature but not enough to guarantee what Diener el al. [8] call “the good life”.

SWB is generally operationalized as a multidimensional construct that refers to perceived satisfaction with one’s life, and a balance between positive and negative affections [11,12,13]. “Hedonic well-being encompasses subjective or emotional well-being, which, in turn, consists of happiness stemming from the components, satisfaction with life and the balance of positive and negative effects.” [14] (p. 1352).

It is also eudaimonic well-being. The term “eudaimonia” is generally translated as well-being. This view compares how well individuals live in terms of their true selves [15] and their attempts to have a meaningful life. In this view, psychological well-being is determined in terms of psychological functioning and personal growth, and it includes how people interact with the environment and adopt a eudemonistic approach to happiness [16].

For Gianetti [17] it is an individual and internal experience, encompassing everything that goes through a person’s mind spontaneously while living their life and interacting with everything around them. Sometimes, their conscious attention turns to particular moments when they realize what they are feeling and thinking, or they reflect on the life they have been leading. Specifically, eudaimonic theories are based on humanistic psychology and argue that not all human desires (even those that result in pleasure) can provide well-being when fulfilled [6,7].

Psychological well-being and social well-being are part of eudaimonic well-being, which results in the inclusion of a range of elements, such as meaning, engagement, purpose in life, positive relationships and personal growth [16,18,19,20].

Psychological well-being can indirectly affect health outcomes via environmental resources that not only improve health-related behaviors but also help individuals cope with stressful events by dampening negative health impacts [21].

Seligman [22] states that a positive state of mind in human beings is associated with protection against colds and flu and, by the same token, a negative state of mind, with a greater risk of developing these diseases. In other words, it may be that highly positive people are less likely to develop cancer.

Henderson and Knight have proposed that these conceptions “should not be treated categorically, nor should they be considered mutually exclusive, but rather that hedonic and eudemonia operate together, in a synergistic way” ([23], p. 201). Therefore, Despite the debate over hedonic versus eudaimonic views of well-being, it appears that well-being is probably best conceived as a multidimensional phenomenon that comprises aspects of both conceptions of well-being [7,24,25,26]. 

As for anxiety, Allen el al. [27] define this as an unpleasant feeling of fear and apprehension, characterized by discomfort derived from anticipating danger, something unknown or strange. They consider that anxiety becomes recognized as a pathology when it reaches an extreme level, disproportionate to the stimulus, which interferes with an individual’s quality of life, the emotional comfort he or she is able to derive from it and a person’s ability to function on a daily basis. In disease classification systems, anxiety disorder is among the five diseases most related to adolescents. Furthermore, according to the World Health Organization (WHO), it is also one of the symptoms of depression and the second leading cause of death by suicide among young people aged 15 to 19 [28].

Spielberg et al. [29] described two types of anxiety: the state of anxiety and the trait of anxiety. According to them, state of anxiety refers to a momentary, transient, tense state, with apprehension raising the activities of the autonomic nervous system, and depending on the perception of the situation, the highest level of state of anxiety is when the situation is perceived as threatening.

The anxiety trait is related to personality and refers to differences in reaction to perceptions of threats with increased state of anxiety. Thus, people who have a high anxiety trait tend to perceive a greater number of dangerous or threatening situations and respond frequently to increased anxiety status [29].

For Souza [30] generalized anxiety disorder (GAD) is defined as a state of intensified concern that can affect several activities of the individual’s life. This can be considered a chronic and recurrent disorder with its psychosomatic symptoms, occurring daily in a period of at least six months.

Ladouceur et al. [31] in the GAD, anxious behavior is driven by internal stimuli, providing information about the state of the organism and external, related to the four senses (vision, hearing, taste and smell) and is the result of learning and thought and memory management. These processes, in turn, are conditioned to the basic knowledge of the subject about himself and the environment. The explanatory knowledge comes from the cognitive representations that the subject constructs from his contact with the environment and by the elaborations carried out in the process of storing knowledge in long-term memory.

For Fava et al. [32] clinically anxious individuals demonstrate a pattern of selective process that operates in order to favor the codification of threatening information. This attention bias was verified in situations of extreme anxiety disorders, which is the case of GAD.

For Castilloa et al. [33] anxiety disorders have symptoms much more intense than day-to-day anxiety. They can appear as exaggerated worries or fears preventing an individual from relaxing; sufferers may feel a continuous sense of tragedy, as if something bad is going to happen; they may be concerned about their health, lack of money, family and work; they may have an exaggerated fear of suffering humiliation and/or bullying; they may feel unable to control their attitudes and thoughts so that these continue or are repeated even if this goes against their will; and sufferers may have a panic syndrome and social panic.

Such disorders (particularly among university students, who are the focus of this research) are aggravated by the advancement of technology, where there is an excess of information, combined with pressure and apprehension about the future, and anxiety or a neurotic need to be perfect. The transition to academic life that these young people face and the excessive demands placed upon them thus generate a series of symptoms and discomfort for young students, which can make it difficult to carry out day-to-day activities [34].

In the context presented, this article aims to interrelate dimensions of the well-being validation instruments proposed by Watson et al. [9] (PANAS) Brazilian version and Mariondo et al. [35] Spanish version, generalized anxiety dimensions proposed by Spitzer et al. [36] (GAD-7) Brazilian version and García-Campayo et al. [37] (2010) Spanish version and state-trait anxiety inventories proposed by Biaggio and Natalício [38] (IDATE) being validation instruments proposed by Watson, et al. [9] (PANAS) Brazilian version and Fonseca and Sepúlveda [39] (2013) Spanish version, using partial least squares structural equation modeling (PLS-SEM).

## 2. Materials and Methods

This research has an exploratory-descriptive character and uses survey-type quantitative analysis as a research strategy. Questionnaires were completed face-to-face with university students between June and August 2019.

Our sample is of the intentional type and was composed of 460 individuals: 277 students from the Federal University of Santa Maria (UFSM) and 183 students from the University of Buenos Aires (UBA). The students were approached by students from the research group, received and signed a free and informed consent form, maintaining the confidentiality of their responses to the research. Even though the subsamples present social and cultural differences between the samples, the results will show that anxiety and well-being do not differ among students, showing that this behavioral illness must be taken seriously and that it affects the population’s well-being.

To achieve the proposed objective, we used a model of structural equations from the adapted steps of Hair Jr. et al. [40] as follows: (a) specification of the structural model; (b) specification of the measurement model; (c) presentation of data; (d) estimation of the path model; (e) evaluation of the measurement model; (f) evaluation of the structural model; and (g) interpretation of the results and final considerations (Table 1).

To confirm the relationships between dimensions, we used PLS-SEM using SmartPLS^®^ software (SmartPLS, Bönningstedt, Germany), version 3.2.9. The Generalized Anxiety Disorder (GAD-7) instrument (also known as General Anxiety Disorder) was developed by Spitzer et al. [36] to assess, diagnose and monitor anxiety disorders, and it was validated by Kroenke et al. [41] This measure consists of seven items, which are arranged on a four-point scale ranging from 0 (never) to 3 (almost every day).

We applied the standardization technique for scales proposed by Lopes [42] in order to classify the subjective well-being of university students, the degree of the trait, state anxiety and generalized anxiety, classified and adapted according to the dimensions of the proposed instruments:$$Ssi=100\times \frac{\left(Sum-Minimum\right)}{\left(Maximum-Minimum\right)}$$
where *Ss**_i_* = standardized score for dimension *i*; *Sum* = sum of valid scores for dimension *i*; *Minimum* = lowest possible scored for dimension *i*; *Maximum* = highest possible scored for dimension *i*.

The scores proposed by the seminal authors of the (i) PANAS, GAD-7 and (ii) IDATE were adapted to a standardized score (*Ss_i_*), as shown in Table 1 above. 

## 3. Analysis and Presentation of Results

Next, the steps to evaluate and confirm the proposed model will be presented.

### 3.1. Structural Model Specification

The path model (see Figure 1), with its respective hypotheses, will be specified first, showing the relationships between the dimensions proposed by the original authors, following the steps proposed by Hair Jr. et al. [40] and Porto [43].

Table 2 presents the four hypotheses that we propose to confirm using the structural model (two of which are subdivided into three sub-hypotheses). 

Table 3 shows latent variables (dimensions) of the model for the PANAS-GAD-7-IDATE scales. Having described the structuring of the model (presented in Table 3), we will now consider specification of the measurement model that represents relationships between the constructs and their corresponding indicator variables.

### 3.2. Specification of the Measurement Model

The second stage of modeling deals with construction of the measurement model (also called the external model), which represents the relationships between dimensions (latent variables) and their corresponding indicator variables [40]. Construction of the measurement model, with latent variables and their respective observed variables (coded), is shown in Table 2.

The measurement diagram and relationship between latent (dimensions) and observed (indicators) variables is presented in Figure 2 and Table 4.

Based on Figure 2, it can be seen that the measurement model has eight parameters (βs), which connect the five latent variables (dimensions) and the 67 observed variables (indicators). According to Hair Jr. et al. [40] a measurement model is specified for the exogenous and endogenous dimensions to have control over which variables describe which constructs. The purpose of the path diagram is to describe the structural equations, as shown in Table 4.

Following structuring of the measurement model, the next step is presentation and preparation of the data in the third stage. 

### 3.3. Presentation of Data

This stage refers to collection and preparation of data for modeling (description of the data-collection procedure). 

### 3.4. PLS Path-Model Estimation

This step concerns estimation of the PLS-SEM model and algorithm (structural equations with partial least squares), starting with the following: recognition of the interrelationships between variables (latent and observable); specification of the structural model (step (a)) and measurement model (step (b)); and consideration of known elements in order to estimate the unknowns. For latter task, the algorithm needs to determine the scores of latent variables that will be the inputs for the partial regression models, resulting in estimates for all relationships [40,43].

The SmartPLS^®^ algorithm was configured for seven criteria. The weighting based on the path was the parameterized system, providing a value for the explanation coefficient (R²) which was more expressive for the endogenous dimensions (predictive variables). The number of iterations was defined as 300, representing the maximum number of iterations to be used to calculate the PLS results. The initial weights for the external indicators were defined as 1.0 [40].

After excluding the indicators with factorial loads below 0.6 and which caused the average extracted variance to be above 0.5, Figure 3 presents the algorithm for the path model.

The PANAS, GAD-7 and IDATE path model presented in Figure 3 shows the cross-factorial loads between indicators, the indicators forming the measurement model and the structural coefficients between dimensions. The values of the determination coefficient R^2^ (which measures the proportion of variance of endogenous dimensions explained by the exogenous dimensions that will be evaluated in step (f)) [40,43] are presented in the dimensions. The systematic evaluation proposed by Hair Jr. et al. [40] is carried out in two stages (stages (e) and (f)), as shown in Table 5.

### 3.5. Evaluation of the Measurement Model

The next step was to evaluate the measurement model using internal consistency composed by Cronbach’s alpha (α), composite reliability (ρ_c_), extracted average variance (AVE) and, finally, evaluation of the discriminant validity (cross-factorial loads, Fornell-Larker criterion and heterotrait-monotrait ratio criterion (HTMT)). The measures are presented below.

#### 3.5.1. Internal Consistency and Composite Reliability

Porto [43] states that evaluation of internal consistency uses Cronbach’s alpha (α) as a traditional criterion, which is an estimate of reliability based on the intercorrelations between observed variables. Composite reliability varies between 0 and 1, just like Cronbach’s alpha. Values between 0.70 and 0.90 are considered good and efficient. Values above 0.90 are not desirable (and are undesirable above 0.95) as this may indicate that respondents have redundancy in their answers [40,43]. AVE values > 0.50 indicate that, on average, the dimension explains more than half of the variance of its indicators [43]. Table 6 presents Cronbach’s alpha, composite reliability and average variance extracted for the PANAS-GAD-7-IDATE model.

As can be seen in Table 6, Cronbach’s alphas and composite reliability for the well-being scale produced values within the proposed measures, that is, between 0.6 and 0.95. Values for the Generalized Anxiety Disorder Scale (GAD-7) (composed of seven indicators distributed within a single dimension) produced values within the conformities. The IDATE-T scale (Dash), with 20 indicators, had a value of 8, and the IDATE-E (State), with 20 indicators, had a value of 11. It was observed that the SAI dimensions presented α > 0.9, and the NAS, SAI and GAD dimensions showed a reliability above 0.90 but not greater than 0.95, which indicates possible redundancy or duplicity in some responses [43]. Discriminant validity is understood as an indicator that shows how much a dimension is truly different from other dimensions due to empirical patterns [43] Analysis of AVE’s indicated possible convergence of the model as the dimensions presented AVE’s greater than 0.50.

#### 3.5.2. Convergent Validity

There are two traditional ways of analyzing convergent validity: using cross-factorial loads (cross-loading) and the Fornell and Larcker [49] criteria, which determine comparison of AVE values for each dimension with Pearson’s correlations. The square roots of AVE’s in columns must be greater than the correlations between dimensions.

The HTMT criterion (Heterotrait-monotrait ratio) is more efficient than the previous ones, since the cross-load and Fornell-Larcker criteria may not identify possible discriminant validity of the model [40,43] Table 7 presents the cross-factorial load values for variables observed in the latent variables. Analysis of discriminant validity according to the Chin [50] criterion (presented in Table 8) makes it possible to verify that the factorial loads of observed variables (indicators) in the original dimensions (latent variables) are greater than the others, verifying that there is validity discrimination according to this criterion.

The second way to analyze discriminant validity involves use of the Fornell-Larcker criterion (F-L), which compares the square root of the AVE with Pearson’s correlations for the other latent variables per line. Porto [43] states that the Fornell-Larcker criterion is based on the idea that one dimension shares more variance with its indicators than with any other dimension (see Table 8). The same table shows the results of the HTMT measurement (heterotrait-monotrait ratio) for the PANAS-GAD-7-IDATE model. For Porto, [43] results close to 1 indicate a lack of discriminant validity. Henseler et al. [51] suggest that values below 0.90 indicate that the model has discriminant validity, and with the bootstrapping method, the upper limit of the 95% confidence interval cannot exceed 1.00 (Table 8).

In Table 8, the PANAS-GAD-7-IDATE model meets the Fornell-Larcker criterion, since the AVE values for the square roots of the dimensions are greater than the correlations between the values in columns. As for the HTMT criterion, the results are below 0.9, indicating possible discriminant validity. They were defined in the bootstrapping parameters, based on a 95% confidence interval. Table 9 presents the values of the intervals of the subsample means for the estimated HTMT.

In Table 9, the “Original Sample (O)” column shows original HTMT values for each combination of dimensions. The “Average Sample (A)” column shows the average values of the 5000 HTMT subsamples calculated by the bootstrap method. The UL_97__.5%_ column shows the upper limit of the confidence interval. It was observed that according to Henseler et al. [51], the values of the upper HTMT limit were below 1.0, attesting to the discriminant validity. The values of the model dimensions do not correlate with other dimensions, so it is assumed that the model may differ.

### 3.6. Evaluation of the Structural Model

This phase focuses on assessing whether the structural model supports the hypotheses, facilitating analysis of the model’s predictive capacity and the relationships between dimensions [43]. Our systematic approach to evaluating the structural model (in line with that of Hair Jr. et al. [40] and Porto [43] is based on the following steps: (i) evaluation of the structural model in terms of its collinearity; (ii) assessment of the level of R^2^; (iii) evaluation of the size of the *f*2 effect; (iv) evaluation of the significance of the betas and confirmation (or not) of the hypotheses about the structural model; and (v) evaluation of predictive relevance (Q^2^) by the blindfolding process.

#### 3.6.1. Collinearity Assessment

Collinearity uses the variance inflation factor (VIF) as a measure, which reveals the degree to which the standard error is increased due to the presence of collinearity [33]. VIF values > 5 indicate a potential collinearity problem. Table 10 presents VIF values for the dimensions of the PANAS-GAD-7-IDATE model.

As can be seen from Table 10, VIF values for the dimensions were all found to be below 5, so it can be said that collinearity did not reach critical levels in terms of the dimensions and is not a problem in terms of estimating the PANAS-GAD-7-IDATE model.

#### 3.6.2. Evaluation of the Coefficient of Determination (R^2^ Value (*p*-Value))

The determination coefficient R^2^ is a measure of the model’s forecasting ability, that is, an evaluation of the portion of the variance of endogenous dimensions which is explained by the structural model [51]. Table 11 presents the R^2^ values and adjusted R^2^ values, and their significance confirmed by the bootstrapping method for the PANAS-GAD-7-IDATE model. 

With Cohen [52] it is suggested that if 0.02 ≤ R^2^≤ 0.075 is classified as a weak effect, 0.075 < R^2^ ≤ 0.19 as a moderate effect and R^2^ > 0.19 as a strong effect, the Positive Affection Scale (EAP) (R^2^ = 0.042) have a coefficient of explanation with a weak and significant effect. The Generalized Anxiety Scale (GAD) (R^2^ = 0.290), Trait Anxiety Inventory (IAT) (R^2^ = 0.369) and State Anxiety Inventory (IAE) (R^2^ = 0.758) feature a very strong effect.

The next step is to determine the significance of the size of the *f*^2^ effect. This measure is used to assess whether there will be a substantial impact on the endogenous dimensions if any dimensions are omitted [40]. Ringle et al. [53] classify this measure as how useful a dimension is for the model. Adapted from Cohen [52] to evaluate the effect, the following is suggested: 0.02 ≤ *f*^2^ ≤ 0.075 (small effect); 0.075 < *f*^2^ ≤ 0.225 (medium effect); and *f*^2^ > 0.225 (great effect).

#### 3.6.3. Evaluation of the *f*^2^ Effect

Table 12 presents the values of the *f*^2^ effect size for the PANAS-GAD-7-IDATE model, with columns showing endogenous dimensions and lines showing exogenous dimensions.

Table 12 shows that the size of the effect of the exogenous TAI dimension on the endogenous SAI dimension (0.825), and the size of the effect of the NAS on the TAI (0.386), and the NAS on the GAP (0.349) are all classified as having a great effect. The size of the effect of the NAS dimension on the endogenous SAI dimension (0.175), and the WBS on the TAI (0.097) are classified as small. In the case of the others (NAS → WBS, WBS → GAD and WBS → SAI), the effects are considered null, and are, therefore, not significant (*p* > 0.05).

The significance of the structural model coefficients (beta values) will now be considered. The model relationships address correlations with the establishment of the null hypothesis (H_0_) as β = 0, and the hypotheses proposed should be rejected when *p* < 0.05, that is, the path coefficient is non-zero [53].

Table 13 presents the values of the relationships between dimensions based on the original sample and the average of the subsamples, standard deviation, *t*-statistic and *p*-values.

In terms of the significance of the beta values, it can be seen that in Table 9, only one relation is below the reference value (Z_5%_ = 1.96), that is, accepting H_0_ and that presented the lowest *f*^2^ (*p* > 0.05), it can be said that the other coefficients proposed for the PANAS-GAD-7-STAI regression model are significant (*p* < 0.05).

Finally, the predictive validity measure Q^2^ (also known as the Stone-Geisser indicator) needs to be applied, which assesses the accuracy of the adjusted model. According to Hair Jr. et al. [40] and Porto [43], in the structural model, Q^2^ values greater than zero indicate the predictive relevance of the path model. The Q^2^ value is obtained using the blindfolding procedure, which is an iterative process that is repeated until each data point is omitted and the model is re-estimated, applied to endogenous dimensions of reflective models [38].

#### 3.6.4. Blindfolding Assessment and Predictive Relevance (Q^2^)

According to Porto [43], the Q^2^ values estimated by the blindfolding process represent a measure of how well the path model can predict the values originally observed. Table 14 shows the values of the predictive relevance measure (Q^2^) for the PANAS-GAD-7-IDATE model.

Table 14 shows the values of Q^2^ based on the total results of the blindfolding procedure, which is the predictive relevance of the model in terms of each endogenous dimension. Based on the results, it can be seen that the model is relevant, since the values of Q^2^ are greater than zero.

### 3.7. Interpretation of Results and Final Considerations

At the end of the internships proposed by Hair Jr. et al. [40] and Porto [43], and the results are interpreted in order to achieve the objective of assessing interrelationships between the well-being scales (PANAS), Generalized Anxiety Scale dimensions (GAD-7) and State-Trait-Anxiety Inventory (STAI) using the structural equation model.

The measurement model was found to present measures of internal consistency: Cronbach’s alpha coefficients and satisfactory reliability. Convergent validity (AVE’s) indicated the convergence of the model, with all constructs presenting AVE’s above 0.5. To analyze the discriminant validity, the HTMT criterion (heterotrait-monotrait ratio) was used, calculated by the bootstrapping procedure, with a parameterization of 5000 subsamples and where values less than 1 were obtained for relationships between the constructs.

The structural model was then evaluated, with the identification of collinearity through the VIF (variance inflation factor) indicator, which presented values below 5 for all dimensions of the model, indicating that the collinearity reached critical levels, presenting no problems for the model estimation.

Calculation of the R^2^ determination coefficient was also applied (a measure of the model’s forecasting capacity), obtaining strong and weak results for predictive capacity. Evaluation of *f*^2^ showed small and large effect sizes. Finally, by carrying out the blindfolding procedure, the Q^2^ predictive validity measure was calculated, which assesses the accuracy of the adjusted model, obtaining values greater than zero with identification of the relevance of the PANAS-GAD-7-STAI model.

Thus, according to the indicators used, it can be inferred that relationships between dimensions of the welfare scale and dimensions of the Generalized Anxiety Scales and State-Trait Anxiety Inventories are supported. The final path diagram for the structural equations is shown in Table 15.

Figure 4 presents the final path model for the combined well-being scale/generalized anxiety/state-trait anxiety inventory (PANAS-GAD-7-STAI).

Corroborating the findings of this research, the following authors, in addition to verifying the hypotheses of this research, brought possible consequences to the relationship between the dimensions.

Andriyani et al. [55] studied the well-being of students in Indonesia as a predictor of trait and state anxiety, as a result of generalized anxiety. The authors suggested that teachers and managers need to develop school programs that can improve students’ well-being, providing new school activities and services that can make students feel comfortable and thrive in schools, preventing school dropout.

Areba et al. [56] applied a survey of students from Somalia and the results presented empirical evidence that positive religious coping can help to increase levels of well-being; and consequently, reduce symptoms of anxiety. Conversely, negative religious coping mechanisms can intensify anxiety symptoms and impair students’ well-being.

Lew et al. [57] researched Chinese university students and reported that the excess of homework, participation in projects and internships lead students to extreme stress, which leads to anxiety and which leads to suicide, as a result, students have little time to participate in extracurricular activities such as relaxing activities and, as a result, promoting well-being. Another reason for the anxiety is due to the adaptation of the students’ new life arrangements, that is, living far from the family.

Machado et al. [58] researched medical students from Pernambuco and as a result of their activities at the University and Environment Hospital, the students obtained, in the great majority, low levels of positive emotions, those who presented high levels of subjective well-being are more satisfied with life, felt more positive emotions, and as a result they are more involved in leisure activities, improving sleep quality, and as a result they presented lower levels of anxiety. Conversely, students who felt more negative emotions tend to be younger and belong to the female sex and have higher anxiety problems.

Based on the analyses, students’ well-being, degree of trait and state anxiety, and degree of generalized anxiety were classified and adapted according to the dimensions of the proposed instruments. Figure 5 shows the welfare situation of students from Buenos Aires and Santa Maria.

In Figure 5, it can be seen that 16.09% of students had low positive affection, and 27.61% had low negative affection. Similarly, with high affection, 16.09% were found to present high positive affection, and 9.61% reported high negative affection.

Comparison of our results with those of Silva and Heleno [59] indicates that where as 13.6% of the students they studied were classified as having low subjective well-being, 14.1% of our students reported this; 72.0% of Silva and Heleno’s students were found to have a moderate level of well-being, compared to 73.7% in our study; and finally, 14.4% of Silva and Heleno’s students had high levels of well-being, and in this study, this reduced 12.17%.

However, a significant difference can be seen between our findings and those of Cunha et al. [60] (who surveyed 174 Portuguese university students) in terms of well-being scores for the positive affect. The Portuguese students had a mean score (standard deviation) of 30.79 (7.627), while those in our research scored 26.06 (7.886). For the negative affect, the Portuguese students had an average score of 15.68 (6.302), and those in our research scored 26.60 (9.900). As for subjective well-being, the Portuguese students had an average score of 15.11 (9.058), and the well-being score for university students in our research was 51.14 (16.091).

Figure 6 shows the state-trait anxiety scores of students from Buenos Aires and Santa Maria. Figure 6 shows that trait anxiety presented as depression in 17.61% of cases, as normal in 7.39% of cases and as anxiety in 75% of university students. In the research by Chaves et al. [61] (applied to 609 university students in Minas Gerais), state anxiety was classified as depression in 7.1% of the sample, as normal in 64.0% and as anxiety in 28.9% of students. As for state anxiety, students from Santa Maria and Buenos Aires were found to present with depression in 13.04% of cases, a normal state in 6.74% of cases and anxiety in 80.22% of cases. As for trait anxiety, in the research by Chaves et al. [61] 8.5% presented as having depression, 60.5% as normal and 31.0% as having anxiety. (There is a significant difference between the students in this research and the Mineiro students, that is, an average difference of 49.22% in the degree of anxiety.)

Compared with the research carried out by Gama et al. [62] with Sergipanos and Paulistas students, whose average degree (standard deviation) of trait anxiety was around 41.4 (9.3) and 43.4 (10.8), respectively, in our research, the students had much higher average scores; the university students in Santa Maria had an average score of 49.62 (11.3), and the university students in Buenos Aires scored even higher with 50.45 (11.7). Figure 7 shows the degree of generalized anxiety, diagnosed by the GAD-7 instrument.

Figure 7 shows that 13.26% of the students can be classified as having a normal level of anxiety, 27.61% as having mild anxiety, 27.61% as having moderate anxiety and 31.74% as having severe anxiety; that is, 86.74% of students had a generalized degree of anxiety. These findings can be compared with a study 1,968 Portuguese university students, carried out by Almeida, [63] who found the following: 47.6% had mild anxiety; 9.7% had a degree of moderate anxiety; and 5.9% had severe anxiety; that is, 63.2% had symptoms of anxiety, and 36.8% had a normal level of anxiety.

## 4. Conclusions

This article aimed to interrelate dimensions of the well-being instruments proposed by Watson et al. [9] (PANAS) with the generalized anxiety dimensions proposed by Spitzer et al. [36] (GAD-7) and the trait anxiety inventories proposed by Biaggio and Natalício [38] (IDATE), in a convenience sample of university students in the city of Buenos Aires and Santa Maria, Brazil.

Research of this nature enables contributions to be presented to the academy confirming or rejecting models of structural equations based on partial least squares, involving scale-based instruments, as well as evaluation of particularities of samples studied in terms of the development of psychological characteristics.

Articles with studies related to well-being and other types of behavioral diseases are being published in the scientific literature. However, the relationship between this dimension and anxiety is still incipient, which allows us to affirm that positive and negative affect are strong predictors of trait and state anxiety, and generalized anxiety, which is the focus of our proposed article.

Our findings indicate strong correlations between the two anxiety instruments, reinforcing the capability of both instruments to assess students’ state of anxiety. The only strong correlation presented was the negative effect scale with the State Anxiety Inventory, whereas the weakest correlation was the positive effect scale with generalized anxiety.

The results of our analyses corroborate other findings in the literature, since high levels of anxiety were found among the university students surveyed. Of the eight hypotheses proposed in our research, only one was not confirmed, but it presented contradictory results to those reported in the literature. The other hypotheses yielded very important findings, which may be confirmed in new research (bearing in mind that confirmation of the beta values of correlations, what is common is just the sign).

### 4.1. Theoretical Implications

In terms of theoretical implications, it was registered that both positive and negative affect prevailed at moderate levels. Positive affect, indicated by students’ enthusiasm, is a satisfactory result, but in contrast, negative affect is a general dimension of anguish and dissatisfaction. This is worrisome as the consequences of negative affect can be linked to mental illness, causing poorer academic performance, thus compromising an individual’s professional future or even causing lack of engagement and discontent with his or her chosen course.

Another significant result of this study is the finding that trait anxiety (which refers to a student’s susceptibility to anxiety) registered highly, which is worrying since it can manifest when students are exposed to conflicting situation. Less than 8% of university students were found to have normal levels of anxiety.

State anxiety, on the other hand, is a fleeting emotion triggered by a certain event, generating upsetting feelings. Results indicated that anxiety had a high incidence rate, followed by depression, and around 7% of students reported normal levels. It is important to note that these findings relate to the student population participating in the present study. With different samples, results may differ as the events that trigger the disorder also differ.

A high degree of generalized anxiety disorder is also highlighted as a theoretical implication (found among about 86% of university students from Buenos Aires/Argentina and Santa Maria/Brazil). Thus, there is a need for future research that contributes to new health measures aimed at minimizing this index, to advance the theoretical contributions of this area.

It is necessary to disseminate these results so that they can be compared with other studies confirming or refuting such implications. In this way, it will be possible to advance research that deals with such subjects among the specific population of university students.

### 4.2. Practical/Managerial Implications

As a practical implication, managers, coordinators and teachers in higher education institutions should recognize the relevance of anxiety during higher education study. Institutional and teaching practices that promote anxiety need to be rethought in order to reduce the rate of mental illness and its consequences (such as disengagement and withdrawal) within academic environments. It is suggested that educational institutions adopt specific measures to minimize such impacts through mental health and quality-of-life programs [64].

Living with anxiety symptoms, such as fatigue, insomnia, difficulty concentrating and irritability, among others, can have a social impact in addition to triggering more serious mental disorders. Mental health is integral to the health of human beings, and early diagnosis brings many benefits, including better well-being. For this, it is necessary to have effective health policies, spanning the period from entry to higher education to entry into the labor market, since the consequences of not doing so will impact on or damage public health.

Finally, one of the limitations of this study which stands out is the sample size. Students from the University of Buenos Aires and the Federal University of Santa Maria (located in the south of Brazil) were evaluated, and it was found that even with a geographical distance of 1060 km and participants coming from countries with different cultures and habits, when it comes to well-being and anxiety, no significant differences were found. It is, therefore, suggested that future studies should involve university students from other regions of Brazil and perhaps other countries in South America, and include other themes related to behavioral diseases among university students.

## Figures and Tables

**Figure 1 ijerph-17-03874-f001:**
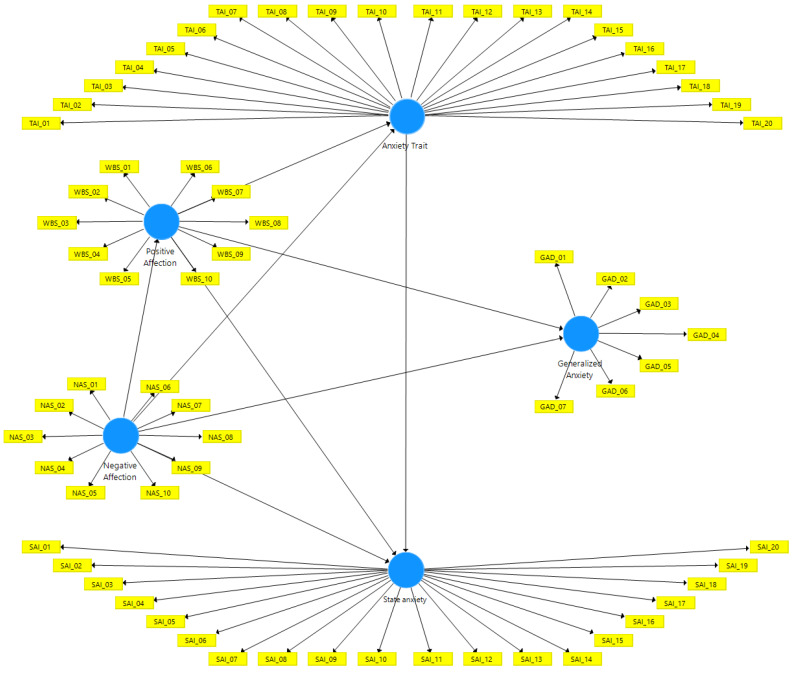
Initial model of the dimensions of the PANAS-GAD-7-IDATE scales. Source: Data research and SmartPLS^®^ software, v. 3.3.2 [44].

**Figure 2 ijerph-17-03874-f002:**
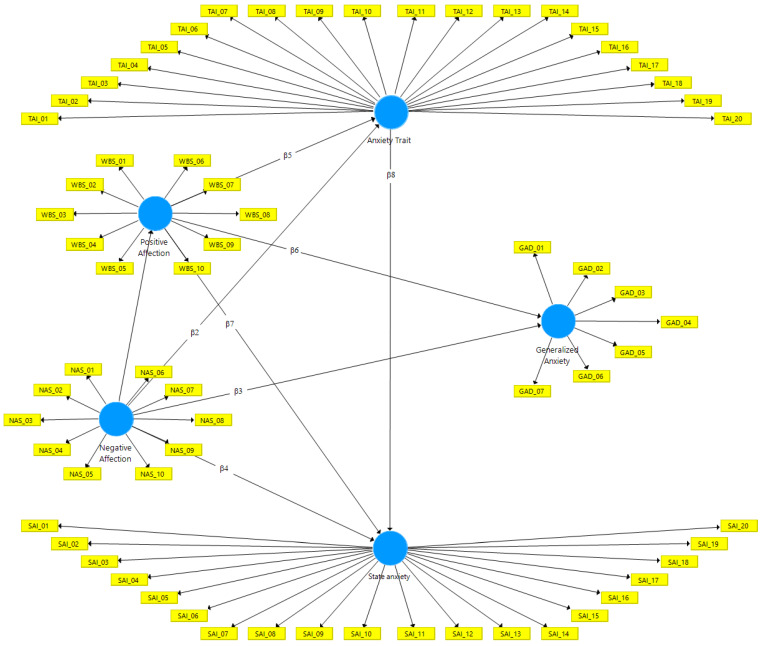
PANAS, GAD-7 and IDATE scale measurement model. Source: SmartPLS^®^ software, v. 3.3.2 [44].

**Figure 3 ijerph-17-03874-f003:**
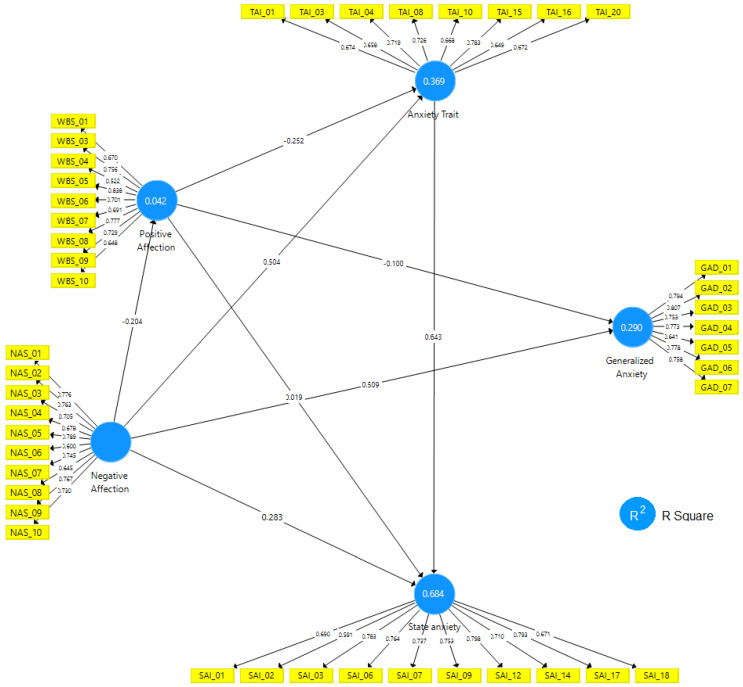
PANAS, GAD-7 and IDATE scale path model Source: SmartPLS^®^ software, v. 3.3.2 [44].

**Figure 4 ijerph-17-03874-f004:**
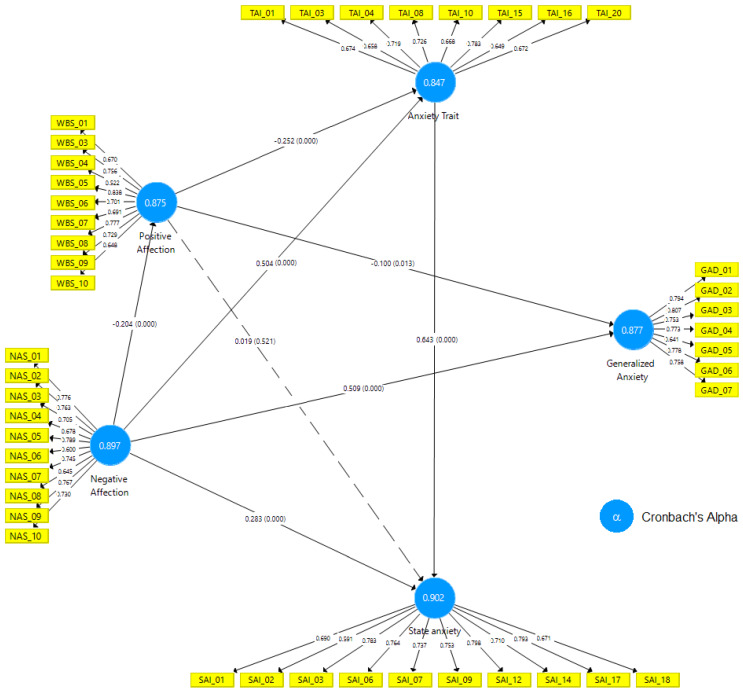
Final path model for the combined well-being scale/generalized anxiety/state-trait anxiety inventory (PANAS-GAD-7-STAI). Source: data research and SmartPLS^®^ software, v. 3.3.2 [44].

**Figure 5 ijerph-17-03874-f005:**
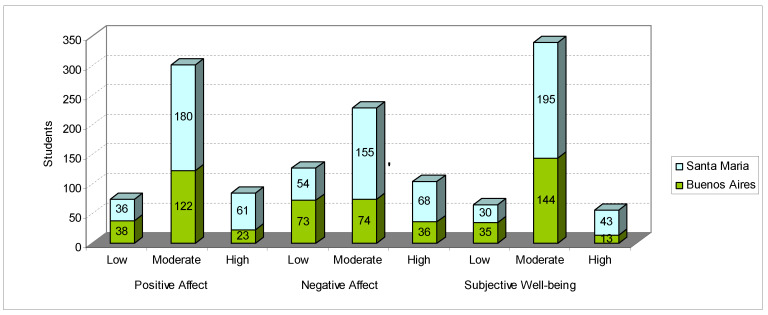
Scale of positive and negative affection of students from Buenos Aires and Santa Maria. Source: survey data.

**Figure 6 ijerph-17-03874-f006:**
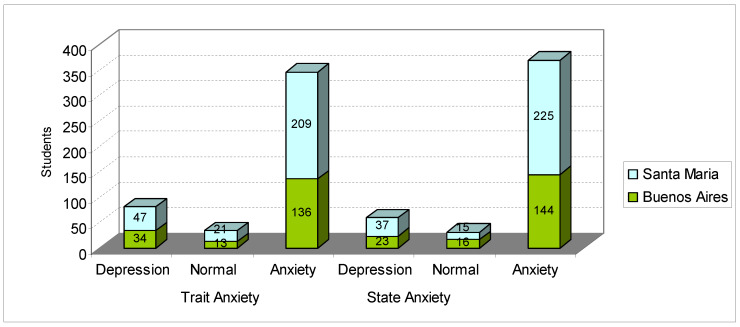
State and trait anxiety inventory of students from Buenos Aires and Santa Maria. Source: survey data.

**Figure 7 ijerph-17-03874-f007:**
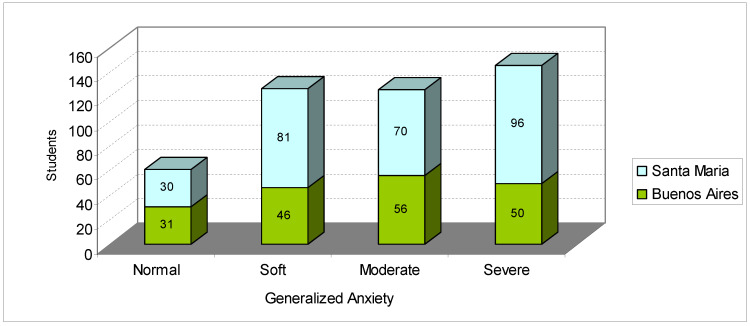
General anxiety disorder levels among students from Buenos Aires and Santa Maria. Source: survey data.

**Table 1 ijerph-17-03874-t001:** Adaptation of scores originally proposed by the authors of these scales, with the standardized score.

Score of the Original Instrument	Proposed Score (Ss_i_)	Classification
PANAS *		
Positive Affection (PA)	0 to 33.33	Low
Negative Affection (NA)		High
Positive and Negative Affection	33.34 to 66.67	Moderate
Positive Affection (PA)	66.68 to 100.00	High
Negative Affection (NA)		Low
Subjective well-being **	−40.00 to −15.00	Low
	−15.01 to 15.00	Moderate
	15.01 to 40.00	High
GAD-7		
1 to 4	0.00 to 15.00	Normal
5 to 9	15.01 to 45.00	Slight
10 to 14	45.01 to 70.00	Moderate
15 to 21	70.01 to 100.00	Severe
IDATE		
20 to 38	0.00 to 30.00	Depression
39 to 42	30.01 to 37.00	Normal
43 to 80	37.01 to 100.00	Anxiety

* Scale has no score proposed by the authors. ** Difference between the sum of positive affection points and the sum of negative affection points. Source: Research data.

**Table 2 ijerph-17-03874-t002:** Proposed hypotheses (H).

Hypothesis	Definition
H_1_	Negative affect directly and negatively influences positive affect. According to Diener and Emmons [45] the relationship between positive and negative affect differs according to the response time, in which the negative relationship is stronger when the reporting of emotions refers to shorter periods than with greater intensities.
H_2_	Negative affect directly and positively influences anxiety: H_2a_: Negative affect directly and positively influences trait anxiety; H_2b_: Negative affect directly and positively influences generalized anxiety; H_2c_: Negative affect directly and positively influences state anxiety. According to Zanon and Hutz [46] the relationship between negative affect and anxiety is positive, since anguish, dissatisfaction, guilt and fear are closely related to danger, threats and daily concerns.
H_3_	Positive affect directly and negatively influences anxiety; H_3a_: Positive affect directly and negatively influences trait anxiety; H_3b_: Positive affect directly and negatively influences generalized anxiety; H_3c_: Positive affect directly and negatively influences state anxiety. The relationship between positive affect and anxiety is reversed as enthusiasm, being excited, optimistic and in a good mood reflect the opposite of fear, dissatisfaction and daily concerns [46].
H_4_	According to Spielberg et al. [29] both types of anxiety (even if conceptualized differently, such as the state of anxiety and the trait of anxiety) differ as the first refers to a momentary, transient state, characterized by tension, apprehension and elevated autonomic nervous system activity, while the second relates to a person’s personality and refers to different reactions to situations perceived as threatening and an increased state of anxiety. Thus, people who have a pronounced anxiety trait tend to perceive a greater number of situations as dangerous or threatening and, consequently, to frequently respond to this increased state of anxiety.

**Table 3 ijerph-17-03874-t003:** Dimensions of the PANAS-GAD-7-IDATE model.

Instrument	Latent Variables	Concepts
Positive and Negative Affect Schedule—PANAS	Positive Affection Scale (WBS)	Reflects how enthusiastic, active and alert a person is [9].
	Negative Affection Scale (NAS)	Reflects anguish and dissatisfaction, including a variety of aversive moods, including anger, guilt, heartbreak and fear.
Generalized Anxiety Scale (Generalized Anxiety Disorder)—GAD-7 (GAD)		A state of exacerbated concern that can affect various activities or events in an individual’s life. With its symptoms (psychiatric and somatic), this can be considered a chronic and recurrent disorder [47].
State-Trait Anxiety Inventory Scale (STAI)	Trait Anxiety Inventory (STAI-T) (TAI)	People who have an anxiety trait tend to perceive a greater number of situations as dangerous or threatening and, consequently, to frequently respond with an increased state of anxiety [48].
	State Anxiety Inventory (STAI-E) (SAI)	Refers to a momentary, transient state, characterized by tension, apprehension and increased autonomic nervous system activity, depending on how a situation is perceived, with a heightened state of anxiety when the situation is perceived as threatening [48].

Source: Watson at al. [9]; Spitzer et al. [36]; Biaggio and Natalicio [38].

**Table 4 ijerph-17-03874-t004:** Diagram of paths for structural equations of the PANAS-GAD-7-IDATE model.

Endogenous Dimensions	=	Exogenous Dimensions	+	Error
WBS	=	β_1_ NAS	+	ε_WBS_
TAI	=	β_2_ NAS *+* β_5_ WBS	+	ε_TAI_
SAI	=	β_4_ NAS + β_7_ WBS + β_8_ TAI	+	ε_SAI_
GAD	=	β_3_ NAS + β_6_ WBS	+	ε_GAD_

Source: survey data based on Hair Jr. et al. [40].

**Table 5 ijerph-17-03874-t005:** Systematic evaluation of PANAS-GAD-7-IDATE results.

**Stage (e) Evaluation of the Measurement Model**
- Cronbach’s alpha (α); - Composite reliability (ρ_c_); - Average variance extracted (AVE); - Cross-factorial loads; - Fornell-Larcker criterion; - Heterotrait-monotrait ratio (HTMT) criteria, confirmed by the bootstrapping method.
**Stage (f) Evaluation of the Structural Model**
- Collinearity assessment (VIF); - Coefficient of determination (R^2^), confirmed by the bootstrapping method; - Effect size (*f^2^*), confirmed by the bootstrapping method; - Conformity of the hypotheses according to the Student’s *t*-test, determined by the bootstrapping method; - Predictive relevance (Q^2^), confirmed by the blindfolding method.

Source: adapted from Hair Jr. et al. [40] and Porto [43].

**Table 6 ijerph-17-03874-t006:** Cronbach’s alpha (α), composite reliability (ρ_c_) and average variance extracted (AVE) for dimensions of the PANAS-GAD-7-IDATE model.

Dimensions	α	ρ_C_	AVE
Negative Affection Scale (NAS)	0.897	0.915	0.521
Positive Affection Scale (WBS)	0.875	0.899	0.502
State Anxiety Inventory (SAI)	0.902	0.920	0.535
Generalized Anxiety (GAD)	0.877	0.905	0.632
Trait Anxiety Inventory (TAI)	0.847	0.882	0.503

Source: SmartPLS^®^ software, v. 3.3.2 [44].

**Table 7 ijerph-17-03874-t007:** Values of the cross-factorial loads for variables observed in the latent variables for the PANAS-GAD-7-IDATE model.

Indicators	Dimensions				
	NAS	WBS	SAI	GAD	TAI
NAS_01	0.776	−0.149	0.526	0.406	0.441
NAS_02	0.763	−0.167	0.502	0.397	0.416
NAS_03	0.705	−0.222	0.419	0.353	0.400
NAS_04	0.678	−0.189	0.417	0.327	0.386
NAS_05	0.789	−0.076	0.477	0.394	0.413
NAS_06	0.600	−0.152	0.364	0.295	0.318
NAS_07	0.745	−0.147	0.482	0.434	0.398
NAS_08	0.645	−0.200	0.386	0.368	0.425
NAS_09	0.767	−0.088	0.510	0.433	0.393
NAS_10	0.730	−0.101	0.479	0.392	0.409
WBS_01	−0.155	0.670	−0.171	−0.156	−0.232
WBS_03	−0.240	0.756	−0.272	−0.215	−0.281
WBS_04	−0.007	0.522	−0.117	−0.084	−0.133
WBS_05	−0.217	0.838	−0.238	−0.186	−0.315
WBS_06	−0.146	0.701	−0.158	−0.153	−0.283
WBS_07	−0.042	0.691	−0.115	−0.031	−0.198
WBS_08	−0.183	0.777	−0.220	−0.139	−0.253
WBS_09	−0.099	0.729	−0.189	−0.131	−0.267
WBS_10	−0.048	0.648	−0.137	−0.111	−0.232
SAI_01	0.365	−0.263	0.690	0.475	0.605
SAI_02	0.376	−0.373	0.591	0.457	0.596
SAI_04	0.501	−0.113	0.783	0.558	0.561
SAI_06	0.573	−0.227	0.764	0.580	0.629
SAI_07	0.465	−0.168	0.737	0.532	0.575
SAI_09	0.439	−0.165	0.753	0.582	0.583
SAI_12	0.545	−0.151	0.798	0.579	0.611
SAI_14	0.455	−0.146	0.710	0.607	0.498
SAI_17	0.511	−0.203	0.793	0.539	0.606
SAI_18	0.383	−0.144	0.671	0.541	0.515
GAD_01	0.422	−0.116	0.626	0.794	0.535
GAD_02	0.420	−0.211	0.622	0.807	0.609
GAD_03	0.367	−0.131	0.527	0.753	0.465
GAD_04	0.391	−0.169	0.555	0.773	0.533
GAD_05	0.268	−0.088	0.443	0.641	0.368
GAD_06	0.472	−0.215	0.591	0.778	0.554
GAD_07	0.427	−0.125	0.570	0.758	0.535
TAI_01	0.354	−0.417	0.470	0.455	0.674
TAI_03	0.419	−0.120	0.564	0.587	0.658
TAI_04	0.380	−0.172	0.539	0.531	0.719
TAI_08	0.471	−0.232	0.608	0.478	0.726
TAI_10	0.269	−0.339	0.430	0.340	0.668
TAI_15	0.478	−0.221	0.609	0.507	0.783
TAI_16	0.272	−0.366	0.437	0.302	0.649
TAI_20	0.386	−0.170	0.685	0.553	0.672

Source: SmartPLS^®^ software, v. 3.3.2 [44].

**Table 8 ijerph-17-03874-t008:** Analysis of discriminant validity using Fornell-Larcker and HTMT criteria with the proposed model.

Dimensions	$\sqrt{AVE}$	Pearson’s Correlation Matrix (F-L)				
		NAS	WBS	SAI	GAD	TAI
NAS	0.722	1.000				
WBS	0.709	−0.204	1.000			
SAI	0.731	0.636	−0.267	1.000		
GAD	0.795	0.529	−0.204	0.725	1.000	
TAI	0.760	0.555	−0.355	0.713	0.685	1.000
		**HTMT**				
NAS						
WBS		0.220				
SAI		0.701	0.287			
GAD		0.626	0.415	0.893		
TAI		0.586	0.215	0.835	0.775	

Source: SmartPLS^®^ software, v. 3.3.2 [44].

**Table 9 ijerph-17-03874-t009:** Confidence interval for HTMT for 5000 subsamples.

Dimension→Dimension	Original Sample (O)	Sample Average (A)	Confidence Interval (CI)	
			IL_2.5%_	UL_97.5%_
WBP→NAS	0.220	0.237	0.172	0.322
SAI→NAS	0.701	0.701	0.633	0.764
SAI→WBS	0.287	0.293	0.205	0.384
GAD→NAS	0.586	0.585	0.504	0.663
GAD→WBS	0.215	0.224	0.143	0.315
GAD→SAI	0.835	0.835	0.785	0.879
TAI→NAS	0.626	0.626	0.544	0.701
TAI→WBS	0.415	0.418	0.324	0.511
TAI→SAI	0.893	0.893	0.859	0.925
TAI→GAD	0.775	0.775	0.714	0.829

IL = inferior limit; UL = upper limit. Source: SmartPLS^®^ software, v. 3.3.2.

**Table 10 ijerph-17-03874-t010:** Values of VIF for the dimensions PANAS-GAD-7-IDATE model.

Dimensions Exogenous	Dimensions Endogenous			
	WBS	SAI	GAD	TAI
NAS	1.000	1.446	1.044	1.044
WBS		1.145	1.044	1.044
TAI		1.586		

Source: SmartPLS^®^ software, v. 3.3.2 [44].

**Table 11 ijerph-17-03874-t011:** R^2^ and R^2^ adjustment coefficient, adjusted for the PANAS-GAD-7-IDATE model.

Dimensions Endogenous	R^2^ (*p*-Value)	R^2^_adjusted_ (*p*-Value)
Positive Affection Scale (WBS)	0.042 (0.037)	0.040 (0.048)
State Anxiety Inventory (SAI)	0.684 (0.000)	0.682 (0.000)
Generalized Anxiety (GAD)	0.290 (0.000)	0.286 (0.000)
Trait Anxiety Inventory (TAI)	0.369 (0.000)	0.367 (0.000)

Source: SmartPLS^®^ software, v. 3.3.2 [44].

**Table 12 ijerph-17-03874-t012:** Size of the *f*^2^ effect for the PANAS-GAD-7-IDATE model.

Exogenous Dimension → Endogenous Dimension	Original Sample (O)	Sample Average (A)	T-Statistics	*p*-Value
NAS → WBS	0.044	0.050	1.944	0.052
NAS → TAI	0.386	0.395	4.929	0.000
NAS → GAP	0.349	0.358	4.958	0.000
NAS → SAI	0.175	0.179	3.931	0.000
WBS → GAD	0.013	0.017	1.103	0.270
WBS → SAI	0.001	0.003	0.215	0.830
WBS → TAI	0.097	0.103	2.734	0.006
TAI → SAI	0.825	0.835	6.416	0.000

Source: SmartPLS^®^ software, v. 3.3.2 [44].

**Table 13 ijerph-17-03874-t013:** Values between latent variables of the PANAS-GAD-7-IDATE model.

Structural Relationship	Hypotheses	Original Sample (O)	St. Deviation (STDEV)	T-Statistic (|O/STDEV|)	*p*-Value	Significance (*p* < 0.05)
NAS → WBS	H_1_	−0.204	0.047	4.353	0.000	Accept
NAS → TAI	H_2a_	0.504	0.036	13.803	0.000	Accept
NAS → GAD	H_2b_	0.509	0.037	13.759	0.000	Accept
NAS → SAI	H_2c_	0.283	0.033	8.504	0.000	Accept
WPS → IAT	H_3a_	−0.252	0.041	6.158	0.000	Accept
WPS → GAD	H_3b_	−0.100	0.041	2.418	0.016	Accept
WPS 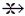 * SAI	H_3c_	0.019	0.029	0.643	0.520	Reject
TAI → SAI	H_4_	0.643	0.033	19.713	0.000	Accept

Source: SmartPLS^®^ software, v. 3.3.2 [44]. * not related.

**Table 14 ijerph-17-03874-t014:** Q^2^ values for the PANAS-GAD-7-IDATE model.

Endogenous Dimensions	SQO	SQE	$Q^{2}=1-\frac{SQE}{SQO}$
Positive Affection Scale	4140.00	4071.32	0.017
State Anxiety Inventory	4600.00	2947.17	0.359
Generalized Anxiety	3220.00	2696.59	0.163
Anxiety Trait Inventory	3680.00	3037.06	0.175

SQO = sum of observed squares; SQE = sum of squares of errors. Source: SmartPLS^®^ software, v. 3.3.2 [44].

**Table 15 ijerph-17-03874-t015:** Final path diagram for structural equations of the PANAS-GAD-7-IDATE model.

Endogenous Dimensions	=	Exogenous Dimensions	+	Error
WBS	=	0.204 NAS	+	ε_WBS_
TAI	=	0.504 NAS − 0.252WBS	+	ε_TAI_
SAI	=	0.283 NAS + 0.643 TAI	+	ε_SAI_
GAD	=	0.509 NAS − 0.100 WBS	+	ε_GAD_

Source: survey data based on Hair Jr. et al. [54].

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
