# Peer review of "Analysis of Well-Being and Anxiety among University Students"

_ijerph, 2020, doi:10.3390/ijerph17113874_

Round 1

Reviewer 1 Report

In the article entitled "Mental Health among university students: an analysis of well-being and anxiety", the authors approach an interesting and relevant objective for the field of knowledge. To this aim, they present a good work well written and with an interesting results and conclusions. For all these reasons I think the work is interesting for publication. However, there are some weaknesses in the work that can help improve it. In the hope that my suggestions will really help to improve it, I will list the problems identified:

Title: I think the tem “mental health” can be confusing. I consider that it is better to use directly anxiety and well-being.

Abstract: I think that the anxiety definition it is not necessary in this section.

-In this section and in the results the authors says “As for generalized anxiety, 399 of the 470 students 37 (86.74%) were found to have generalized anxiety”. With the current wording seems to be a large clinical sample. It is needed to the rewritten.

Introduction:

Thera are only a 17.87% of references for woks published in the last five years. It is important to actualize the conceptual framework.

It is not clear why the authors selected this constructs. In most of the introduction they present well being and this importance. But it is not the same with anxiety and there is no explanation of why it is selected between al variables linked wit well being.

Materials and Methods:

This is the most problematic section. When I read this section I had many doubts. The description of the sample (centers, type of career, academic year, etc.) is not clear. This makes it impossible to know the capacity of generalization of the study. How the samples were carried out (complete classes, individually, by a single applicant or several, etc.) is also not clear, making it difficult to replicate them. In addition, there are other fundamental aspects such as informed consent and approval by an ethics committee that have been omitted.  All of this information needs to be expanded so that it can be published.

Results:

This section is very weel done, but I do not understand why to compare between centers. I think the results are enough complex and interesting without this analysis.

It is important to report on the number of lost data and their treatment. With such a adjusted sample this point is very relevant as it can affect the results obtained.

Author Response

Dear Reviewer

We would like to thank the reviewer for taking the time and effort to read and review our manuscript. We are grateful for the constructive comments provided, and we have revised the manuscript in accordance with the suggestions made by the reviewers and are pleased to submit a revised version of our manuscript.

Reviewer 1#

In the article entitled "Mental Health among university students: an analysis of well-being and anxiety", the authors approach an interesting and relevant objective for the field of knowledge. To this aim, they present a good work well written and with an interesting results and conclusions. For all these reasons I think the work is interesting for publication. However, there are some weaknesses in the work that can help improve it. In the hope that my suggestions will really help to improve it, I will list the problems identified.

Title: I think the “mental health” can be confusing. I consider that it is better to use directly anxiety and well-being.

Response: We thank the reviewer for this important observation. The title has been changed as suggested: “Analysis of well-being and anxiety among university students”

Abstract: I think that the anxiety definition it is not necessary in this section.

Response: Thanks! We agree with this suggestion and rewrote the abstract.

-In this section and in the results the authors says “As for generalized anxiety, 399 of the 470 students 37 (86.74%) were found to have generalized anxiety”. With the current wording seems to be a large clinical sample. It is needed to the rewritten.

Response:  Thanks for the observation. We rewritten this part. In terms of the subjective well-being of students, 14.13% were found to have a low rating. 86.74% were found to have generalized anxiety; 75.00% had trait anxiety, and 80.22% had state anxiety.

Introduction:

Thera are only a 17.87% of references for woks published in the last five years. It is important to actualize the conceptual framework. It is not clear why the authors selected this constructs. In most of the introduction they present well being and this importance. But it is not the same with anxiety and there is no explanation of why it is selected between al variables linked wit well being.

Response: We followed Reviewer 1's suggestion and included other, more up-to-date references, combined with new findings on the subject (well-being and anxiety among university students).

Materials and Methods:

This is the most problematic section. When I read this section I had many doubts. The description of the sample (centers, type of career, academic year, etc.) is not clear. This makes it impossible to know the capacity of generalization of the study. How the samples were carried out (complete classes, individually, by a single applicant or several, etc.) is also not clear, making it difficult to replicate them. In addition, there are other fundamental aspects such as informed consent and approval by an ethics committee that have been omitted. All of this information needs to be expanded so that it can be published.

Response: Thank you for this important observation. Made us improve substantially what was pointed out. We made the necessary adjustments and highlighted in the text. We include the question of the translation of research instruments into the Spanish language as well as the ethical aspects of research.

Compliance with Ethical Standards - Ethics approval and consent to participate; This research did not receive any specific grant from funding agencies in the public, commercial, or not-for-profit sectors. The Regional Committee for Research Ethics of the Federal University of Santa Maria approved the study protocol (project number 054104 - CAEE: 30859 720.8.000.5346- CONEP). The study was performed in accordance with relevant guidelines and regulations.

Results:

This section is very well done, but I do not understand why to compare between centers. I think the results are enough complex and interesting without this analysis. It is important to report on the number of lost data and their treatment. With such a adjusted sample this point is very relevant as it can affect the results obtained.

Response:  Our goal was to make this part clear to the reader. Also, we had no data lost due to the instruments having been applied digitally, where the respondents indicated the consent to publish the information that was collected. The choice of the sample containing Brazilians and Argentines is because the researchers sought significant evidence on the cultural differences of these countries in terms of anxiety, which showed us that anxiety is not a particular phenomenon.

Reviewer 2 Report

The author(s) presented an interesting topic and, in general, a paper with publication potential. 

The paper demonstrate an adequate understanding of the relevant literature in the well-being field and cite an appropriate range of literature sources, however, we found some assymetric investment in the main topics, in particular, the trait and state anxiety and generalized anxiety should be theoretical developed. We think that could be done a deep effort to update the literature reviewed, in particular, in the topics inter-relations, that could enable to support the empirical analysis and could enrich the main results discussion. Some assurance of exhaustivity in the literature search, in particular in the university population, could be relevant. 

The links between the theoretical and empirical component and the empirical part of the paper, in particular, the elements presented about used instruments (versions/adaptations) and the measurement model should be verified and improved. In the Method part a lot of relevant information should be added, in particular, about the instruments used and procedures, because the data collection occurred in two different countries with different language and cultural specificities. The procedures - of instruments adaption and data collection (in two different countries) - must be explained with more detail. 

The paper' main results were well presented, in graphic terms and content,  but the discussion and conclusion could not add, in our opinion, a real add value (or outstanding contribution) to this scientific domain state-of-art. Additionally, the (practical) implications were not well developed and not take in consideration the sample social-demografic characteristics or the Universities analyzed and respective countries cultural specificities. Conclusions and implications were formulated in a too generic way and this is a strong limitation for the publication in an international journal.

Author Response

Reviewer #2

Comments and Suggestions for Authors

The author(s) presented an interesting topic and, in general, a paper with publication potential.

The paper demonstrate an adequate understanding of the relevant literature in the well-being field and cite an appropriate range of literature sources, however, we found some assymetric investment in the main topics, in particular, the trait and state anxiety and generalized anxiety should be theoretical developed. We think that could be done a deep effort to update the literature reviewed, in particular, in the topics inter-relations, that could enable to support the empirical analysis and could enrich the main results discussion. Some assurance of exhaustivity in the literature search, in particular in the university population, could be relevant.

Acknowledgments: We would like to thank the reviewer for taking the time and effort to read and review our manuscript. We are grateful for the constructive comments provided, and we have revised the manuscript in accordance with the suggestions made by the reviewers and are pleased to submit a revised version of our manuscript.

Response: We followed suggestion and made a significant improvement in the literature consulted regarding personality traits and generalized anxiety, seeking to contextualize its consequences.

The links between the theoretical and empirical component and the empirical part of the paper, in particular, the elements presented about used instruments (versions/adaptations) and the measurement model should be verified and improved. In the Method part a lot of relevant information should be added, in particular, about the instruments used and procedures, because the data collection occurred in two different countries with different language and cultural specificities. The procedures - of instruments adaption and data collection (in two different countries) - must be explained with more detail.

Response: We thank you for this important observation and we find the suggestions pertinent. We tried to improve the text, mainly in the methodology with explanation of how the translation of the research instrument was carried out.

Reviewer 3 Report

L = Line number

  1. The manuscript focuses on a relevant social theme and it is very well structured, especially regarding the methodology (section 2 and parts of section 3). It also includes relevant and up-to-date references on the themes and methodological strategies used.

  1. Table 1 – further specify * and ** in the table, in L162 -163.

  1. In Table 2, it would be clearer if there was, for example, a dashed line indicating the separation of Hypotheses H1 to H4.

  1. In Figure 2, is missing

  1. Pages 9, 10, 11 - Standardize the nomenclature:

Average variance extracted (AVE)

extracted average variance (VME)

  1. L325-326-327 – Verify

"The only dimension that stands out with a coefficient of explanation, with a very strong effect, is the State Anxiety Inventory (IAE) (R2 = 0.758)."

  1. L413-414-415-416-417 – the results from Silva and Heleno (2012) are mixed up:

 "Comparison of our results with those of Silva and Heleno50 indicates that whereas 14.13% of the students they studied were classified as having low subjective well-being, 13.6% of our students reported this; 73.70% of Silva and Heleno’s students were found to have a moderate level of well-being, compared to 72.0% in our study; and finally, 12.17% of Silva and Heleno’s students had high levels of well-being, and in this study, this rose to 14.4%."

  1. L432-433-434 – Review

"(There is a significant difference between the students in this research and the Mineiro students, that is, an average difference of 47.7% in the degree of anxiety.)"

  1. Reference 56 is not cited in the text.

Kashdan, T.B., Biswas-Diener, R., & King, L.A. (2008). Reconsidering happiness: The costs of distinguishing  between hedonics and eudaimonia. Journal of Positive Psychology, 3, 219–233.

  1. Minor reviews

L12

L32-33 - abstract

  1. In section 3.7 (L413 to L452), the authors compare their results with other studies. They should emphasize that the comparisons are merely illustrative, because the studies are not directly comparable. In addition, other studies do not always adopt the same scales. However, they correctly recognize that: "It is important to note that these findings relate to the student population participating in the present study. With different samples, results may differ as the events that trigger the disorder also differ." (L489-490-491).

Author Response

Reviewer #3

The paper' main results were well presented, in graphic terms and content, but the discussion and conclusion could not add, in our opinion, a real add value (or outstanding contribution) to this scientific domain state-of-art. Additionally, the (practical) implications were not well developed and not take in consideration the sample social-demografic characteristics or the Universities analyzed and respective countries cultural specificities. Conclusions and implications were formulated in a too generic way and this is a strong limitation for the publication in an international journal.

Resposta:

Thanks for this suggestion. This was also a concern of the reviewer 2. We rewrote several parts of the text to make this issue clearer. Also, we show in the body of work the fact that the present research shows that, even in students with different cultures, the anxiety phenomenon is present and certainly influences their day-to-day or their well-being.

Comments and Suggestions for Authors
L = Line number

The manuscript focuses on a relevant social theme and it is very well structured, especially regarding the methodology (section 2 and parts of section 3). It also includes relevant and up-to-date references on the themes and methodological strategies used.

Response: Thank you so much. This observation is very important to us.

Table 1 – further specify * and ** in the table, in L162 -163.

Response: We apologize for this error. We made all the necessary adjustments.

In Table 2, it would be clearer if there was, for example, a dashed line indicating the separation of Hypotheses H1 to H4.

Response: Thank you for this observation and suggestion. It will show this topic in a more didactic way to the reader. We include the dashed line to separate the hypotheses.

In Figure 2, is missing

Response: We have included Table 4 in addition to Figure 2 in the text to elucidate any doubts and comply with what was pointed.

Pages 9, 10, 11 - Standardize the nomenclature:
Average variance extracted (AVE)
extracted average variance (VME)

Response: we standardize a single abbreviation for the entire text (AVE), as pointed.

L325-326-327 – Verify
"The only dimension that stands out with a coefficient of explanation, with a very strong effect, is the State Anxiety Inventory (IAE) (R2 = 0.758)."

Response: We rewrote the settings that were previously pointed. Thanks!

L413-414-415-416-417 – the results from Silva and Heleno (2012) are mixed up: "Comparison of our results with those of Silva and Heleno50 indicates that whereas 14.13% of the students they studied were classified as having low subjective well-being, 13.6% of our students reported this; 73.70% of Silva and Heleno’s students were found to have a moderate level of well-being, compared to 72.0% in our study; and finally, 12.17% of Silva and Heleno’s students had high levels of well-being, and in this study, this rose to 14.4%."

Response: Thanks! The information previously mentioned has been corrected.

L432-433-434 – Review
"(There is a significant difference between the students in this research and the Mineiro students, that is, an average difference of 47.7% in the degree of anxiety.)"

Response: The value indicated was corrected and the percentage equal to 49.22% was duly inserted.

Reference 56 is not cited in the text.
Kashdan, T.B., Biswas-Diener, R., & King, L.A. (2008). Reconsidering happiness: The costs of distinguishing between hedonics and eudaimonia. Journal of Positive Psychology, 3, 219–233.

Response: Some references were excluded and, later, new references were inserted in the article, which ended up changing its numbering.

Minor reviews
L12
L32-33 - abstract

In section 3.7 (L413 to L452), the authors compare their results with other studies. They should emphasize that the comparisons are merely illustrative, because the studies are not directly comparable. In addition, other studies do not always adopt the same scales. However, they correctly recognize that: "It is important to note that these findings relate to the student population participating in the present study. With different samples, results may differ as the events that trigger the disorder also differ." (L489-490-491).

Response: We take care to include new (recent) findings pointed out by Reviewer, to highlight the relevance of the theme, as well as, what should be done in terms of studies related to well-being and anxiety in study environments.

We hope that these changes meet your expectations, and look forward to your consideration.

Acknowledgment

We are grateful to anonymous reviewers for the tips contributions and recommendations. All comments were very helpful, and we believe that by addressing the comments and suggestions, our revised manuscript was significantly improved.

Round 2

Reviewer 2 Report

Author(s) have improved selectively, but significantly, the article.  But, one important topic was still missing. In the "method" when the authors said: "the instruments were translated and adapted to the Spanish language, with the help of professors from the University of Buenos Aires...", additional and specific information, must be added about the process. About the "How". The instruments are crucial for this kind of research validity,. So this is, for us, a critical factor for the scientific publication of this research.

The article should be verified, also, in the tables, figures (e.g., use of colours) in accordance with the journal guidelines. English language and style are fine/minor spell check required. Format of the references should be verified in accordance with the journal guidelines (e.g., use of capital letters in some references).

Author Response

We would like to thank the reviewers and editor for taking the time and effort to read and review our manuscript entitled “Analysis of well-being and anxiety among university students”. We are grateful for the constructive comments provided, and we have revised the manuscript in accordance with the suggestions made by the reviewers and are pleased to submit a revised version of our manuscript.

In the following pages, we have detailed all our responses (highlighted in green) to each of the specific comments from the reviewer and described the changes that have been made to our manuscript.

We hope that these changes meet your expectations, and look forward to your consideration.

Sincerely,

#Reviewer 2

Author(s) have improved selectively, but significantly, the article.  But, one important topic was still missing. In the "method" when the authors said: "the instruments were translated and adapted to the Spanish language, with the help of professors from the University of Buenos Aires...", additional and specific information, must be added about the process. About the "How". The instruments are crucial for this kind of research validity,. So this is, for us, a critical factor for the scientific publication of this research.

Response: Thanks for the important observation. We made the corrections to the text and added new references to validate the translated versions of the languages. The changes were highlighted in yellow in the main text.

I inserted three new authors, regarding the validation of the three instruments in Spanish.

“In the context presented, this article aims to interrelate dimensions of the well-being validation instruments proposed by Watson, Clark and Tellegen9 (PANAS) Brazilian version and Mariondo et al. 35 Spanish version, generalized anxiety dimensions proposed by Spitzer et al.36 (GAD-7) Brazilian version and García-Campayo et al.37 (2010) Spanish version and state-trait anxiety inventories proposed by Biaggio and Natalício38 (IDATE) being validation instruments proposed by Watson, Clark and Tellegen9 (PANAS) Brazilian version and Fonseca and Sepúlveda39 (2013) Spanish version, using partial least squares structural equation modeling (PLS-SEM).”

- Mariondo, M; Palma, P.; Medrano, L. A.; Murillo, P. Adaptación de la Escala de Afectividad Positiva y Negativa (PANAS) a la población de adultos de la ciudad de Córdoba: análisis psicométricos preliminares. Univ. Psychol. 2011; 11(1): 187-196.

- García-Campayo, J.; Zamorano, E.; Ruiz, M. A.; Pardo, A.; Pérez-Páramo, M.; López-Gómez, V.; Freire, V. O.; Rejas, J. Cultural adaptation into Spanish of the generalized anxiety disorder-7 (GAD-7) scale as a scrrening tool. Heath Qual Life Outcomes. 2010; 8(8): jan.

- Biaggio, A. M. B.; Natalicio, L. Manual para o Inventário de Ansiedade Traço-Estado (IDATE). Rio de Janeiro: Centro Editor de Psicologia Aplicada-CEPA; 1979.

The article should be verified, also, in the tables, figures (e.g., use of colours) in accordance with the journal guidelines. English language and style are fine/minor spell check required. Format of the references should be verified in accordance with the journal guidelines (e.g., use of capital letters in some references).

Response: Thank you for this important suggestion. We have remade all the references, tables and figures for the journal style. The article was reviewed by a native English speaker.

We are grateful to anonymous reviewers for the contributions and recommendations. All comments were very helpful, and we believe that by addressing the comments and suggestions, our revised manuscript was significantly improved.